# Evaluation of a bioaerosol sampler for indoor environmental surveillance of Severe Acute Respiratory Syndrome Coronavirus 2

**Patrick Finn Horve** [1], **Leslie Dietz**[1], **Dale Northcutt**[1,2], **Jason Stenson**[1,2], **Kevin Van Den Wymelenberg** [1,2,3]*

**1** Biology and the Built Environment Center, University of Oregon, Eugene, OR, United States of America,
**2** Energy Studies in Buildings Laboratory, University of Oregon, Eugene, OR, United States of America,
**3** Institute for Health and the Built Environment, University of Oregon, Portland, OR, United States of America

* kevinvdw@uoregon.edu

**Data Availability Statement:** All relevant data can be accessed at https://github.com/BioBE/AerosolSense-LabTests.

**Funding:** Van Den Wymelenberg serves as a scientific advisor to EnviralTech, a company that

## Abstract

The worldwide spread of Severe Acute Respiratory Syndrome Coronavirus 2 (SARS-CoV-2) has ubiquitously impacted many aspects of life. As vaccines continue to be manufactured and administered, limiting the spread of SARS-CoV-2 will rely more heavily on the early identification of contagious individuals occupying reopened and increasingly populated indoor environments. In this study, we investigated the utility of an impaction-based bioaerosol sampling system with multiple nucleic acid collection media. Heat-inactivated SARS-CoV-2 was utilized to perform bench-scale, short-range aerosol, and room-scale aerosol experiments. Through bench-scale experiments, AerosolSense Capture Media (ACM) and nylon flocked swabs were identified as the highest utility media. In room-scale aerosol experiments, consistent detection of aerosol SARS-CoV-2 was achieved at an estimated aerosol concentration equal to or greater than 0.089 genome copies per liter of room air (gc/L) when air was sampled for eight hours or more at less than one air change per hour (ACH). Shorter sampling periods (75 minutes) yielded consistent detection at ~31.8 gc/L of room air and intermittent detection down to ~0.318 gc/L at (at both 1 and 6 ACH). These results support further exploration in real-world testing scenarios and suggest the utility of indoor aerosol surveillance as an effective risk mitigation strategy in occupied buildings.

## Introduction

Since the onset of the Severe Acute Respiratory Syndrome Coronavirus 2 (SARS-CoV-2) pandemic, all aspects of typical life and society have been altered to limit the spread of Coronavirus Disease 2019 (COVID-19). As knowledge has accrued and a better understanding of the spread of SARS-CoV-2 in the environment has developed, aerosols have been increasingly implicated in the spread of COVID-19 [1–5]. Aerosols are found in a variety of sizes ranging from sub-micron to larger particles (>100 μm) and can stay suspended in indoor air for minutes to hours [4–6]. Built environments, including offices, schools, gyms, places of worship, cars, public transportation, and other human-inhabited indoor spaces [7], can be especially

conducts viral environmental surveillance, including in senior care facilities. EnviralTech did not play a role in the study design, data collection and analysis, decision to publish, or preparation of the manuscript. This research was funded by Thermo Fisher Scientific under award number 4133V1. The funder provided the university with support in the form of salaries for all authors, equipment, and reagents but did not have any additional role in the study design, data collection and analysis, decision to publish, or preparation of the manuscript. However, per contractual obligations, the funder had the right to review the final manuscript for confidential information prior to submission. The specific roles of these authors are articulated in the 'author contributions' section.

**Competing interests:** Van Den Wymelenberg has a company called Duktile through which he provides healthy building consulting, including consulting related to viral pathogens, and he serves as a scientific advisor to EnviralTech, a company that conducts viral surface environmental surveillance, including in senior care facilities. This does not alter our adherence to PLOS ONE policies on sharing data and materials

vulnerable to aerosol-based pathogen transmission [8, 9]. The presence of indoor biocontaminants can be exacerbated by poor indoor air exchange rates, low proportions of outside air, poor filtration efficiency, and low indoor humidity, leading to prolonged exposure to potentially infectious aerosols [6, 10–12]. While many healthcare-focused built environments utilize increased ventilation, outside air fraction, and enhanced filtration to limit the potential for aerosols to spread pathogens [13], most building infrastructure does not implement these enhanced ventilation practices and has limited capability to improve ventilation and filtration [14].

As vaccine rollout increases [15–17], reopening will continue and indoor spaces will become more crowded. In addition, new variants are emerging, some of which may escape vaccine-induced immune responses [18–22]. Given this landscape, environmental surveillance for SARS-CoV-2 and identification of asymptomatic individuals shedding SARS-CoV-2 indoors, will become essential to provide early warning of potential outbreaks within a building [23] and to focus further diagnostic testing and guide increased environmental risk reduction strategies.

Previously, high flow aerosol sampling has been employed for biohazard surveillance to combat bioterrorism and accidental release of high-level biohazards [24]. We evaluated a novel aerosol sampling system for its ability to identify virus-containing aerosols within the built environment. Utilizing heat-inactivated SARS-CoV-2, we tested the utility for multiple media to capture and release viral RNA as well as a prototype aerosol sampler in bench-scale and room-scale virus-containing aerosol capture trials.

## Materials and methods

### *Bench series 1*: Capture media testing

Five unique media types including flocked swabs (Typenex, Catalog #SW0202) [25–29], cotton swabs (Puritan, Catalog #25-8061WC) [25, 26, 28, 29], glass fiber filters (Millipore, Catalog #HAWP04700) [30–33], FTA cards (Whatman, Catalog #29277432) [34–37], and Aerosol-Sense Capture Media (ACM) that have demonstrated previous success in nucleic acid isolation were tested within an aerosol sampling platform [24] (Thermo Fisher Scientific, Catalog #2900-AA). SARS-CoV-2 deposited by the Centers for Disease Control and Prevention (CDC) and obtained through BEI Resources, NIAID, NIH (SARS-Related Coronavirus 2, Isolate USA-WA1/2020, NR-52281) were cultured using Vero E6 cells (ATCC CRL-1586) for three to four days in Dulbecco's Minimum Essential Medium (DMEM, ATCC, Catalog #30–2002) at 5% $CO_2$. Viral supernatants were inactivated through heat inactivation and 254 nm ultraviolet (UV) light inactivation. Viral supernatants in 1.5 mL screw-cap tubes were incubated at 65˚C for 20 minutes, following previously established protocols [38]. Tubes were then transferred to chilled Armor Beads (Lab Armor, LLC) to end the inactivation reaction. For UV-inactivation, 1.5 mL screw-cap tubes with viral supernatants were exposed to UV light (266 nm) for 10 minutes. Subsequently, viral supernatant from these treated cultures were serially diluted (10-fold) in viral transport medium (Rocky Mountain Biologicals, Catalog #VTM-CHT) and inoculated onto Vero E6 cells and incubated for 1 hour at 5% $CO_2$ to facilitate infection. Cells were then overlaid with a diffusion limiting agent (0.75% methylcellulose, 1X DMEM, and 2% fetal bovine serum). After incubation for four days, the absence of plaques was observed to confirm inactivation. Heat-inactivated viral stocks were selected due to superior genome stability compared to those inactivated through ultraviolet radiation [38]. The number of viral genomes in each supernatant was determined through absolute quantification using the Charité/Berlin (WHO) protocol primer and probe panel [39], and artificial RNA standards targeting the SARS-CoV-2 RdRP (ORF1ab) and E gene regions [38]. The stock solution was found to have a concentration of genomes per μL.

To assess the ability of each media to uptake and release SARS-CoV-2, five replicates of each media type were incubated for four hours in 10-fold serial dilutions of viral supernatants with concentrations ranging from $3.2 \cdot 10^6$ to $3.2 \cdot 10^1$ genomes/μL diluted with viral transport medium. Following a 4 hour incubation, each media was transferred to a 5 mL tube with a very small amount of the media trapped in the snap-top lid, centrifuged for three minutes at 1,500 x g to remove all liquid from the media, and the media was discarded after centrifugation. An equal volume of DNA/RNA Shield (Zymo Research, Catalog #R1200) as recovered supernatant was added to each tube and stored at 4˚C until RNA extraction. RNA was extracted from 400 μL of the recovered supernatant using the Quick-DNA/RNA Viral Mag-bead kit (Zymo Research, Catalog #R2141) following the manufacturer protocol. The success of the extraction was determined using *Escherichia coli* virus MS2 spike-in [40]. The presence of SARS-CoV-2 RNA was detected using the TaqPath COVID-19 Combo Kit (Thermo Fisher Scientific, Catalog #A47814) targeting the N, S, and ORF1ab (RdRP) gene regions. Reaction mixtures contained 5 μL TaqPath 1-Step Multiplex Mastermix without ROX (Thermo Fisher Scientific, Catalog #A28521), 9 μL nuclease free water (Invitrogen, Catalog #4387936), 1 μL COVID-19 Real Time PCR Assay Multiplex Mix (Thermo Fisher Scientific, Catalog #A47814), and 5 μL of extracted RNA. Thermocycling was performed using a QuantStudio5 (Applied Biosystems, Catalog #A28140) using the following conditions: 25˚C for 2 minutes, 53˚C for 10 minutes, 95˚C for 2 minutes, and 40 cycles of 95˚C for 3 seconds and 60˚C for 30 seconds. A sample was considered positive for the presence of SARS-CoV-2 RNA if amplification was observed in two out of the three genome targets, following the FDA Emergency-Use Authorization guidelines in the assay instructions for use [41]. All reaction plates included an extraction control and a PCR no template control (NTC) to assess potential contamination during RNA extraction and PCR preparation respectively. A positive result in either of these controls invalidated the full plate and required a re-extraction (positive extraction control) or a rerun of the qRT-PCR reaction (positive PCR NTC). All work surfaces were decontaminated using a 10% bleach solution and RNaseAway (Thermo Scientific, Catalog #7003)

### *Bench series 2 and 3*: Aerosol sample collection

A prototype model of the AerosolSense Air Sampler (Thermo Fisher Scientific, Catalog #2900-AA) was utilized to capture SARS-CoV-2 containing aerosols (Fig 1A). Air was sampled at a rate of 200 L/min through a vertical collection pipe and impacted onto the collection media. The AerosolSense Air Sampler is designed to collect aerosolized particles with a diameter between 0.1–15μm [42]. The collection media was held by a removable cartridge, allowing the cartridge to be removed and decontaminated between sampling events (Fig 1A). The collection media was tested for the presence of SARS-CoV-2 using the same molecular methods described above.

Aerosolization took place inside an 818-GB glovebox (Plas Labs, Catalog #818-GB) with an inside height of 104 cm, an inside depth of 71 cm, an inside width of 66 cm, and an internal volume of 489 L (Fig 1B). Previous investigations at healthcare facilities guided the aerosolized concentrations of SARS-CoV-2 tested [1, 5, 43]. Aerosol concentrations in real-world scenarios are typically measured by genome copies captured per liter (gc/L) of air collected. Previous work has demonstrated that captured concentrations of SARS-CoV-2 in built environments range from 3 gc/L [1] to 94 gc/L [5] of room air. First, the AerosolSense system was tested across a large range of potential indoor aerosol doses (*bench series 2*) and then tested at a narrower range of doses (*bench series 3*) against potential scenarios more typical of indoor environments. The doses tested in *bench series 2* included 0.0032 gc/L (1.6 genome copies total), 0.032 gc/L (16 genome copies total), 0.32 gc/L (160 genome copies total), 3.2 gc/L (1600

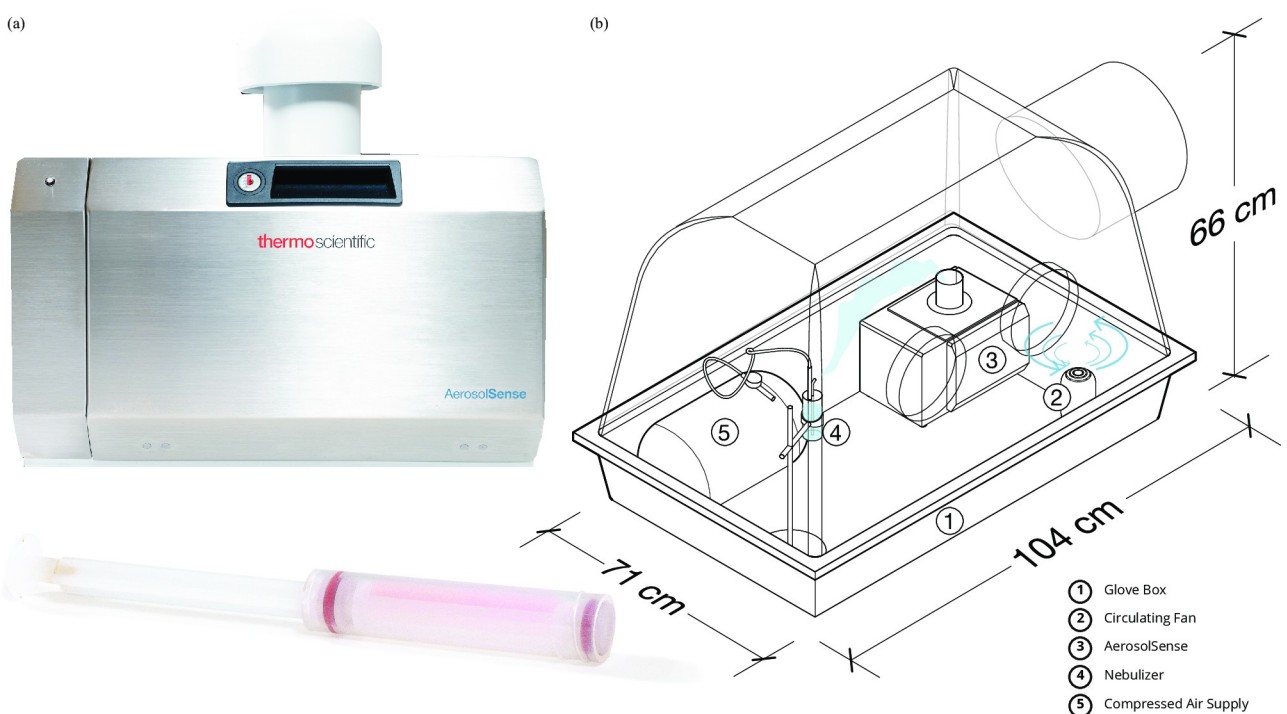

**Fig 1.** (a) AerosolSense sampler and media cartridge; republished from [75] under a CC BY license, with permission from Thermo Fisher Scientific, original copyright 2021. (b) Glove box layout for bench-scale aerosol sample collection.

genome copies total), 32 gc/L (16,000 genome copies total), 320 gc/L (160,000 genome copies total), and 3,200 gc/L (1,600,000 genome copies total). The doses in *bench series 3* included 0.032 gc/L (16 genome copies total), 32 gc/L (16,000 genome copies total), and 320 gc/L (160,000 genome copies total). All doses are reported as dosed gc/L of air volume in the glove box irrespective of time (Table 1). Viral supernatants were nebulized using the Harvard Apparatus Aerosol Nebulizer (Harvard Apparatus, Catalog #73–1963). Aerosol collection proceeded as follows: the sampler and nebulizer were turned on simultaneously, virus was aerosolized for two minutes (1 mL of viral supernatant aerosolized), and the AerosolSense was allowed to run for three additional minutes, resulting in ~2 volumetric air captures to occur in the glove box. Following aerosolization, the capture media was removed from the glove box, 500 μL of DNA/RNA Shield and 500 μL of 1X phosphate buffered saline (PBS) were added to the capture media, incubated for 10 minutes, vortexed for 5 seconds, then centrifuged as described above. RNA isolation, SARS-CoV-2 quantification, and interpretation were performed as described above.

The potential for real-world factors to impact the air sampler's utility in detecting SARS-CoV-2 RNA was also investigated. Particularly, we investigated the impact of 1) glove

**Table 1. Dosed aerosol concentrations and predicted total genomes aerosolized in bench-scale and room-scale aerosolization experiments.**

| Bench-Scale Aerosols *(bench series 2+3)* | | Room-Scale Aerosols | |
|---|---|---|---|
| **Dosed Aerosol Concentration** | **Dosed Genome Copies** | **Dosed Aerosol Concentration** | **Dosed Genome Copies** |
| 0.0032 gc/L | 1–2 genome copies | 3.2 gc/L | 89,728 genome copies |
| 0.032 gc/L | 16 genome copies | 32.0 gc/L | 897,280 genome copies |
| 0.32 gc/L | 160 genome copies | 320 gc/L | 8,972,800 genome copies |
| 3.2 gc/L | 1,600 genome copies | 3,200 gc/L | 89,728,000 genome copies |
| 32.0 gc/L | 16,000 genome copies | 32,000 gc/L | 897,280,000 genome copies |
| 320 gc/L | 160,000 genome copies | | |

box relative humidity (RH) levels, 2) delayed sample processing, and 3) high dust loads. The glove box naturally maintained a consistent relative humidity between 40%-60% RH (mid RH condition) based upon ambient laboratory conditions during testing. A high RH condition (>70%) was achieved by misting water into the enclosed glove box and a low RH condition (<30%) was achieved through the use of lithium chloride [44]. The temperature and RH were monitored throughout the course of all experiments using HOBO UX100-011A data loggers. Delayed processing samples were transported directly from the glove box to airtight plastic boxes. Replicate samples were held for periods of 24, 48, and 72 hours, then processed as described previously. Additionally, the delayed sample airtight plastic boxes were maintained at three different RH conditions to match the conditions in which the aerosol sampling took place (high, mid, or low RH). The temperature and RH within these containment boxes were monitored using HOBO UX100-011A data loggers. The impact of household dust on SARS-CoV-2 aerosol detection was tested by rolling the collection media in vacuumed dust collected from a home in Eugene, Oregon. The household had no history of COVID-19 infection and four aliquots of the dust tested negative for the presence of SARS-CoV-2 prior to use in experiments.

## Room-scale aerosol sample collection

To investigate the ability of the AerosolSense sampler to perform under real-world conditions, heat-inactivated SARS-CoV-2 was aerosolized within a rapid deployable module (RDM, Western Shelter Systems, Eugene) (Fig 2). The interior volume of the RDM was 28,040 L. The viral doses aerosolized into the room were 3.2 gc/L, 32 gc/L, 320 gc/L, 3,200 gc/L, and 32,000 gc/L. These concentrations are reported as dosed gc/L of air volume in the RDM irrespective of time (Table 1). Virus was aerosolized using three 4-jet Blaustein Atomizing Modules (CH Technologies) with a flow rate of 16 L/min at 50 psi for each nebulizer. Nebulization occurred for 60 minutes for all 75-minute sampling durations and 240 minutes for all sampling durations of 8-hours or more. At the beginning of each sampling day, negative control sampling took place

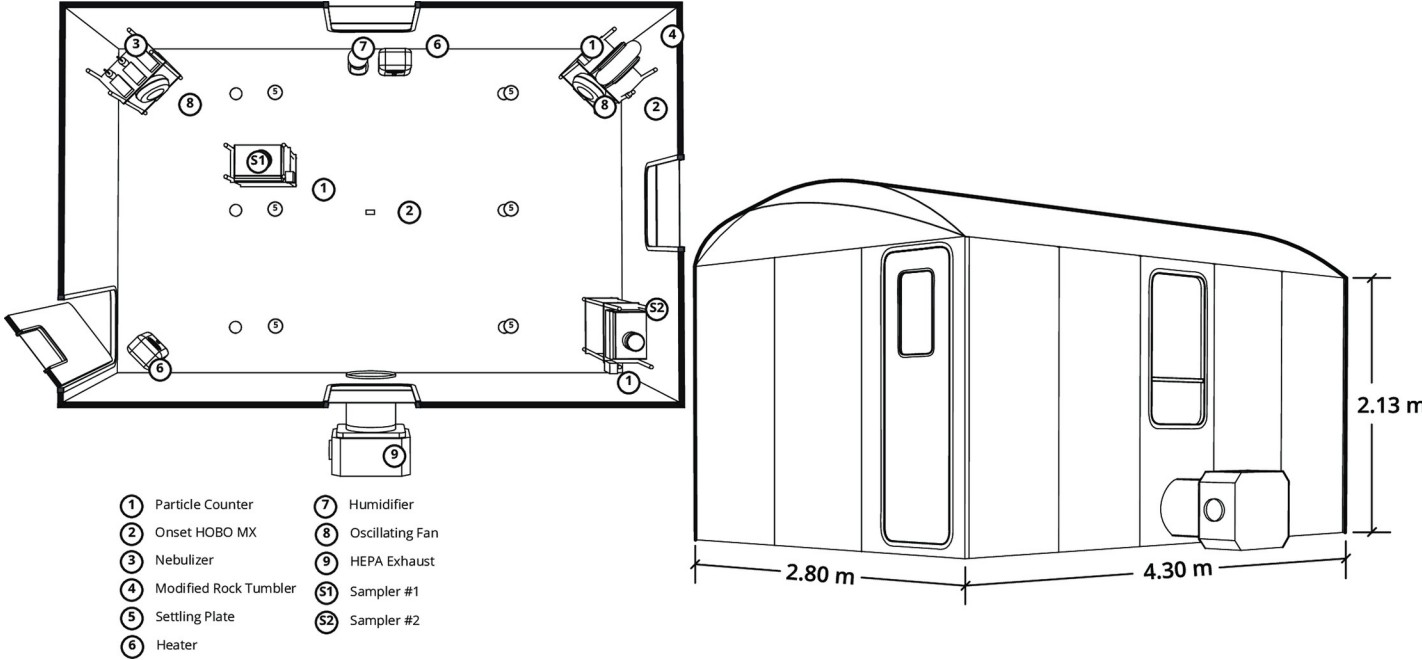

**Fig 2. Layout of environment, AerosolSense samplers, and dimensions for room-scale aerosol testing.**

through the aerosolization of VTM to confirm the environment was negative for SARS-CoV-2 contamination. While the total quantity of aerosolized SARS-CoV-2 dosed into the RDM air volume resulted in the nominal aerosol doses listed in Table 1, the rate of nebulization over time, room air movement, room air exchange rates during sampling, and air infiltration resulted in estimated concentrations that were substantially lower than reported in Table 1 at any given point in time. To address this in relation to the observed results, a model taking into account the number of genome copies aerosolized, nebulization rate and duration, room volume and air exchange rate, deposition in the room, and imperfect nebulization and viral capture was utilized to estimate the aerosol SARS-CoV-2 concentration throughout the course of the sampling time (Fig 3). Based on this model, aerosol SARS-CoV-2 concentrations were estimated to be ~9.95% and ~2.75% of the nominal concentration described in Table 1 for the 75-minute and >8 hour room-scale experiments respectively (Table 2).

## Nebulization aerosol particle characterization

The Harvard Apparatus Aerosol Nebulizer emits particles ranging from <0.1 μm to 10.5 μm, based upon the literature available from the manufacturer. There was no formal data available on the 4-jet Blaustein Atomizing Modules (BLAM nebulizer). In order to better contextualize the detection capability of the AerosolSense sampler and its potential application in real-world scenarios, a characterization of the particle output of the nebulizer was performed. A single BLAM nebulizer was placed in a sealed Purifier Logic+ Class II, Type A2 biosafety cabinet (LabConco, Catalog #302420001) with an Aerotrak 9306-V2 particle counters recording particle counts in 6 size bins (0.3–1 μm, 1–2.5 μm, 2.5–3 μm, 3–5 μm, 5.0–10 μm and 10 μm). VTM was nebulized for 20 minutes in order to achieve steady state particle concentrations. A summary of the overall particle output can be found in Table 3.

For all 75-minute experiments, interior temperature was maintained at 22°C +/- 4°C with two portable electric resistance fan heaters and RH was maintained at 50% +/- 10% using a single portable humidifier. Additional air circulation was provided in the space from two oscillating fans each moving 24,975 liters per minute. Air exchange rate was controlled and maintained at either one or six air changes per hour (ACH) through timed operation of HEPA filtered exhaust air removal from the RDM with make-up air via infiltration. Temperature and RH were monitored and recorded using two Onset HOBO MX1102A data loggers. Exhaust air flow rate was confirmed using an Omega HHF92A CFM Master II anemometer. For high dust (interference) experiments, household vacuum dust was continuously emitted to room air via a modified single drum rotary rock tumbler (Harbor Freight Tools, Calabasas) placed over one of the oscillating fans. Three TSI Aerotrak 9306-V2 particle counters recorded particle counts in six bin sizes (0.3–1 μm, 1–2.5 μm, 2.5–3 μm, 3–5 μm, 5.0–10 μm and 10–25 μm) during all experiments. After each experimental trial, air in the RDM was filtered at ~30 ACH for 10 + minutes using a CL-ACXE1200 HVAC system (Western Shelter Systems) fitted with an additional in-line HEPA filter. This unit was also used to maintain temperature in the RDM in between each experimental trial. For all long duration experiments, viral supernatants were diluted to 24 mL of total liquid in each nebulizer to allow for a longer nebulization duration (240 minutes). Additionally, ambient temperature was not maintained for longer trials (range of 25°C-5°C) and the air exchange rate was maintained at approximately 0.8 ACH.

## Ethics statement

The research described did not require institutional review board approval. However, bench-scale capture media testing protocols were reviewed and approved by the University of Oregon Institutional Biosafety Committee (Registration #2020–19). All protocols relating to the room-

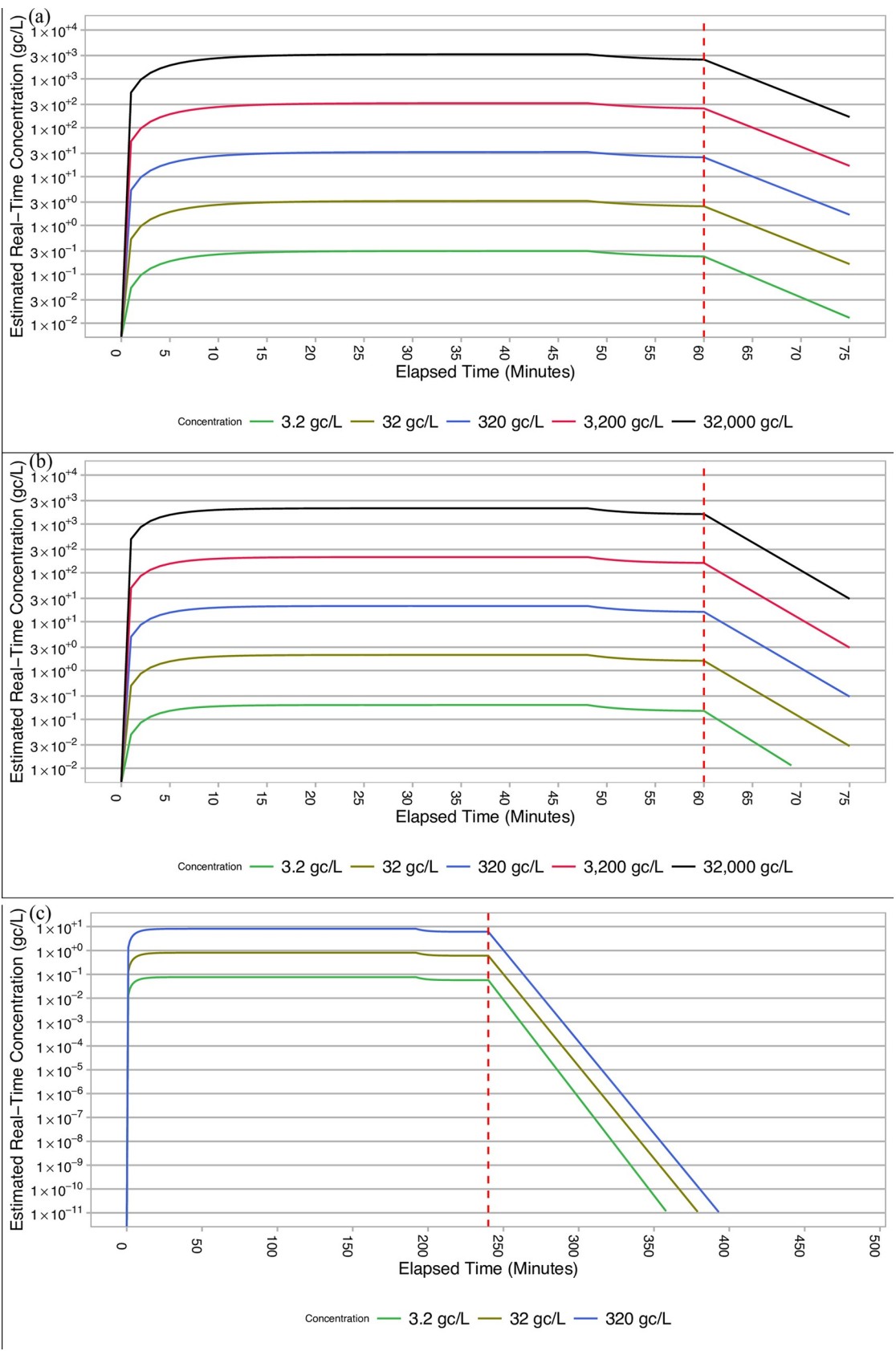

**Fig 3.** Estimated number of aerosolized genome copies throughout the course of 75-minute trials at (a) 1 ACH (b) 6ACH and (c) overnight room-scale aerosol sampling trials. The red dashed line represents the point when the nebulizers were estimated to no longer be nebulizing any supernatant containing heat-inactivated SARS-CoV-2.

scale aerosol captures trials that took place in the RDM were reviewed and approved by Advarra IBC (Protocol #202000110). Advarra IBC is an authorized external IBC for the University of Oregon and is registered with the National Institute of Health (NIH). Data and analysis are available at https://github.com/BioBE/AerosolSense-LabTests.

### Statistical analyses

One-way ANOVA with Tukey's HSD was used to compare the results of the different capture media. Student's t-tests were used to compare the detected viral load between differential environmental conditions and treatments. Differences were considered significant with $P < 0.05$. All statistical analyses were conducted with the statistical programming language, R [46].

## Results

### Capture media testing—qualitative assessment

Each capture media was qualitatively evaluated and scored based on a variety of characteristics using a 5-point Likert scale during the laboratory testing (Table 4), with "1" acting as the lowest score and "5" acting as the highest score [47]. The stability of each media was assessed following incubation, vortexing, and centrifugation. ACM, cotton swabs, and flocked swabs demonstrated no change throughout this process, thus earning a five on the Likert scale. FTA cards and glass fiber filters become soft and fell apart throughout the process, earning them lower marks. All liquid could be centrifuged from both the ACM and flocked swabs and earned a score of five. The eluate was also easily retrieved from the cotton swabs, but the wooden shaft on the model tested led to some retention of liquid. When the FTA cards and glass fiber filters remained intact, liquid retrieval was comparable with that of other materials. However, media disintegration during processing led to difficulty retrieving the liquid. Due to the lack of stability, the FTA card and glass fiber filters were given lower ease of use scores. The cotton swab did not have a preset tip breakpoint, complicating processing for this capture media. This resulted in a decreased score in the ease-of-use category. ACM and flocked swabs were easily integrated into the sample processing workflow and received the top score for ease of use. Lastly, when frozen, all media types performed similarly and received top scores in this category.

**Table 2. Estimated steady-state aerosol concentrations.**

| 1 ACH | | 6 ACH | | Overnight (0.8 ACH) | |
|---|---|---|---|---|---|
| **Nominal Aerosol Concentration (gc/L)** | **Estimated Steady-State Genome Concentration (gc/L)** | **Nominal Aerosol Concentration (gc/L)** | **Estimated Steady-State Genome** | **Nominal Aerosol Concentration (gc/L)** | **Estimated Steady-State** |
| 3.2 | 0.318 | 3.2 | 0.197 | 3.2 | 0.089 |
| 32 | 3.18 | 32 | 1.97 | 32 | 0.89 |
| 320 | 31.8 | 320 | 19.7 | 320 | 8.9 |
| 3,200 | 318 | 3,200 | 197 | 3,200 | 89 |
| 32,000 | 3,180 | 32,000 | 1,970 | 32,000 | 890 |

**Table 3. Characterization of nebulizer aerosolized particle distribution.**

| Harvard Apparatus Aerosol Nebulizer | | BLAM Nebulizer | |
| --- | --- | --- | --- |
| *Particle Bin Size* | *% of Total Particles* | *Particle Bin Size* | *% of Total Particles* |
| 0–1.22 μm | 52.0% | 0.3–1.0 μm | 91.7% |
| 1.22–2.4 μm | 9.5% | 1.0–2.5 μm | 7.9% |
| 2.4–3.04 μm | 6.8% | 2.5–3.0 μm | 0.22% |
| 3.04–4.97 μm | 23.9% | 3.0–5.0 μm | 0.17% |
| 4.97–8.19 μm | 7.5% | 5.0–10 μm | 0.05% |
| 8.19–10.5 μm | 0.2% | 10–25 μm | <0.01 |

Data describing the particle distribution of the Harvard Apparatus Aerosol Nebulizer (left) was adapted from product literature [45].

## *Bench series 1*: Capture media testing—quantitative assessment

The choice of capture media was found to have a significant impact on the observed cycle threshold (Ct) values ($F_{5, 405}$ = 3.722, $P<0.05$, Fig 3A). No significant differences were observed in Ct values between the viral supernatants and liquid eluted from ACM, cotton swabs, flocked swabs, or FTA cards (Fig 3B) in "*bench series 1*" experiments. A significant difference was found in the observed Ct values (higher, less abundance) between the viral supernatants and the liquid eluted from glass fiber filters (Fig 3B). Due to the significant difference in observed Ct values, the glass fiber filter was removed from future experiments. Because there was no significant difference in the observed Ct values between ACM, FTA cards, flocked swabs, and cotton swabs, the qualitative survey was used to determine the capture media that would continue on to the "*bench series 2 and 3*" aerosol experiments. The difficulty in working with FTA cards in the processing pipeline led to its removal from consideration. Lastly, although similar results were observed between the ACM, flocked swabs, and cotton swabs, the lack of breakpoint and slight variation in elution caused by the wooden handle of the cotton swab, it was decided to perform the "*bench series 2 and 3*" aerosol experiments with the ACM and flocked swabs only (Table 5).

## *Bench series 2 and 3*: Aerosol testing

Throughout all bench-scale aerosol tests, the chamber temperature was maintained at 23.8˚C ± 1.32, 25.5˚C ± 2.07˚C, and 24.1˚C ± 0.427˚C for the low, mid, and high RH conditions respectively. A full outline of environmental summary statistics can be found in the supplementary data. Similar to the previous media tests, capture media was not found to significantly impact ($P = 0.89$) observed Ct values across seven decades of concentration and in potential scenario testing ($P > 0.05$), but Ct values did demonstrate a significant relationship to aerosolized viral genome copies ($P < 0.001$). SARS-CoV-2 RNA was detected at least 50% of the time at a concentration of 0.32 gc/L (dosed into glove box) and 3.2 gc/L by the

**Table 4. Qualitative characteristics of each capture media tested and ratings of four factors based upon common laboratory activities.**

| | Stability | Eluate Retrieval | Ease of Use | Stability during Freeze | Totals |
| --- | --- | --- | --- | --- | --- |
| **ACM** | 5 | 5 | 5 | 5 | **20** |
| **Cotton Swab** | 5 | 4 | 3 | 5 | **17** |
| **Flocked Swab** | 5 | 5 | 5 | 5 | **20** |
| **FTA Card** | 2 | 4 | 3 | 5 | **14** |
| **Glass Fiber Filter** | 2 | 3 | 2 | 5 | **12** |

**Table 5. Percent of media tested positive for the presence of SARS-CoV-2 RNA expressed as genome copied dosed per L of chamber air.**

| AerosolSense Capture Media | | Flocked Swabs | |
|---|---|---|---|
| **Dosed Aerosol Concentration** | **Percentage Positive** | **Dosed Aerosol Concentration** | **Percentage Positive** |
| 0.0032 | 0% | 0.0032 | 0% |
| 0.032 | 33.3% | 0.032 | 33.3% |
| 0.32 | 50.0% | 0.32 | 33.3% |
| 3.2 | 66.6% | 3.2 | 25.0% |
| 32 | 100% | 32 | 33.3% |
| 320 | 87.5% | 320 | 66.6% |
| 3,200 | 100% | 3,200 | 100% |

Samples were considered positive if two out of the three genomic targets (N, S, ORF1ab) returned a Ct value ≤35.

ACM and flocked swabs respectively (Fig 4B). SARS-CoV-2 RNA was never detected at doses below 0.032 gc/L (Fig 4B). The ability to capture and detect SARS-CoV-2 RNA was not significantly impacted by aerosol capture at low or high humidity levels across all doses tested (Fig 5B). Similarly, delayed processing across all RH levels and up to three days post-collection did not demonstrate statistically significant results (Fig 5C).

## Room-scale aerosol testing

Once again, the capture media was not found to have a statistically significant impact on the measured Ct value ($P = 0.661$), and the presence of a higher dust load and increased distance from the nebulization source were not found to significantly impact the detection of the aerosolized virus ($P = 0.308$ and $P = 0.622$ respectively). However, increased aerosolized viral load was associated with decreased Ct value on the capture media ($P < 0.001$) and the length of the aerosol capture time was found to significantly decrease the measured Ct value ($P < 0.001$). For the shorter sampling period (75 minutes), aerosolized SARS-CoV-2 RNA was detected at least 50% of the time at 320 gc/L dosed into RDM air (Table 6) while SARS-CoV-2 RNA was detected 100% of the time when the sampling time was increased to greater than 8 hours (at all three doses, 3.2 gc/L, 32 gc/L, 320 gc/L) (Table 6).

## Discussion

Overall, we sought to optimize and evaluate the potential utility of the AerosolSense sampler as a surveillance tool to identify COVID-19 outbreaks when reopening built environments. To this end, this study evaluated the ability of multiple types of media to capture, stabilize, and integrate into a standard SARS-CoV-2 molecular diagnostic workflow [48]. The media tested had previously demonstrated promise for the collection of viral RNA [24–37], either as a media typically associated with prolonged stability of the collected nucleic acids, previous use in aerosolized virus collection, or use in clinical specimen collection. In bench-scale trials, there was a significant link between media that were capable of reliably releasing stored viral supernatants (ACM, flocked swab, cotton swab) and higher concordance to viral supernatant controls. Additionally, media that were found to be easier to handle (ACM, flocked swab, cotton swab) were also found to have results more closely following those of the supernatant controls (Fig 3). In order for environmental surveillance to occur and not place unnecessary strain on capable molecular laboratories, it is essential that the media selection for aerosol sampling fit into existing molecular workflows. Based on these criteria, it was decided that ACM, which most readily released the captured supernatant; and flocked swabs, one of the most common

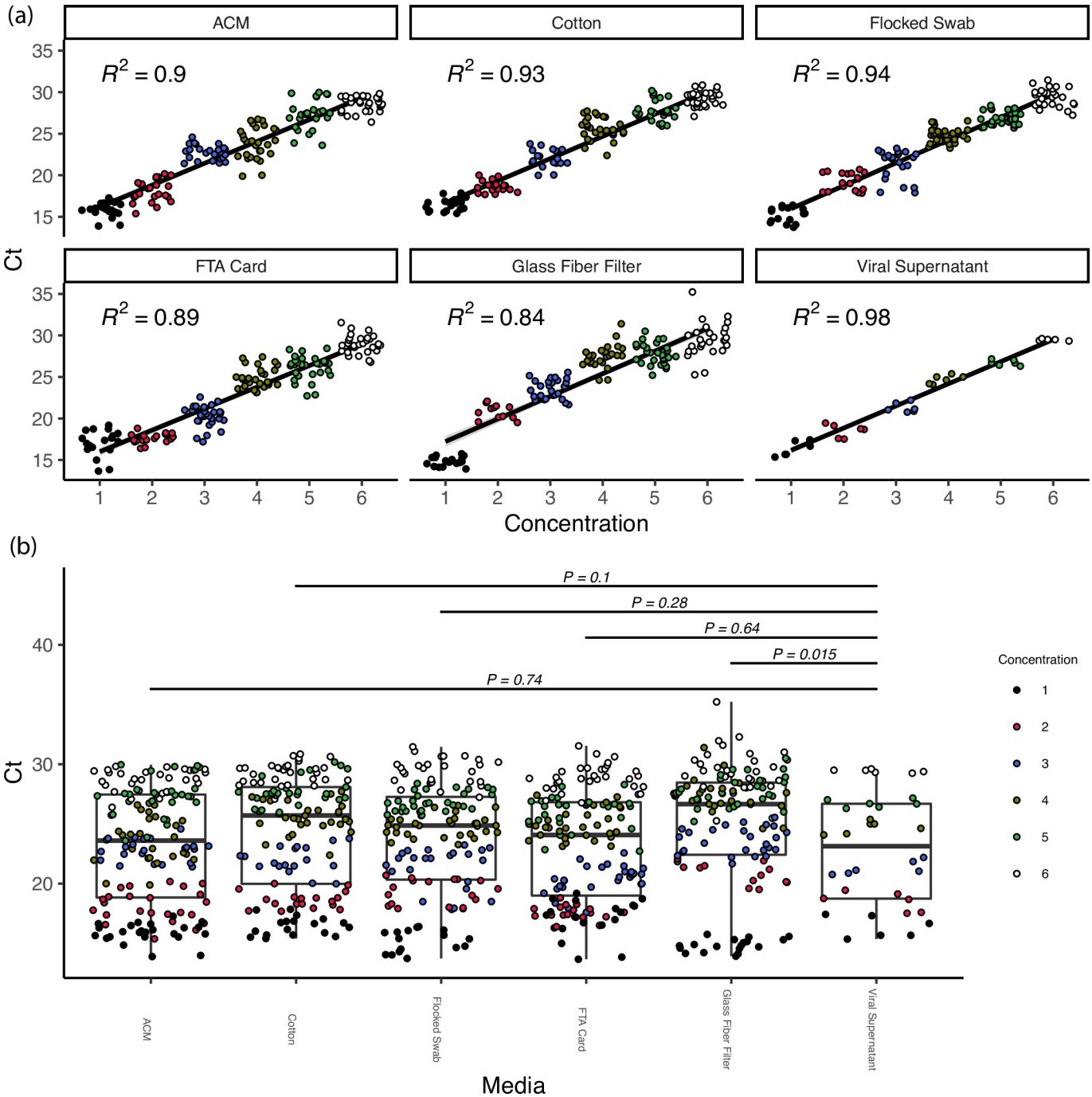

**Fig 4. *Bench series 1* results.** Capture testing media results. (a) Measured Ct values across the serial dilution curve, ranging from $3.2 \cdot 10^6$ genome copies per μL (concentration 1) to $3.2 \cdot 10^1$ genome copies per μL (concentration 6) (b) Boxplots of measured Ct values of liquid eluted from capture media across all concentrations tested. Concentrations are the same as in (a).

media for clinical diagnostic specimen collection [25–27, 29, 49], would proceed to additional bench-scale and room-scale aerosol trials. Similar media had previously demonstrated success in environmental viral sampling [24, 50, 51].

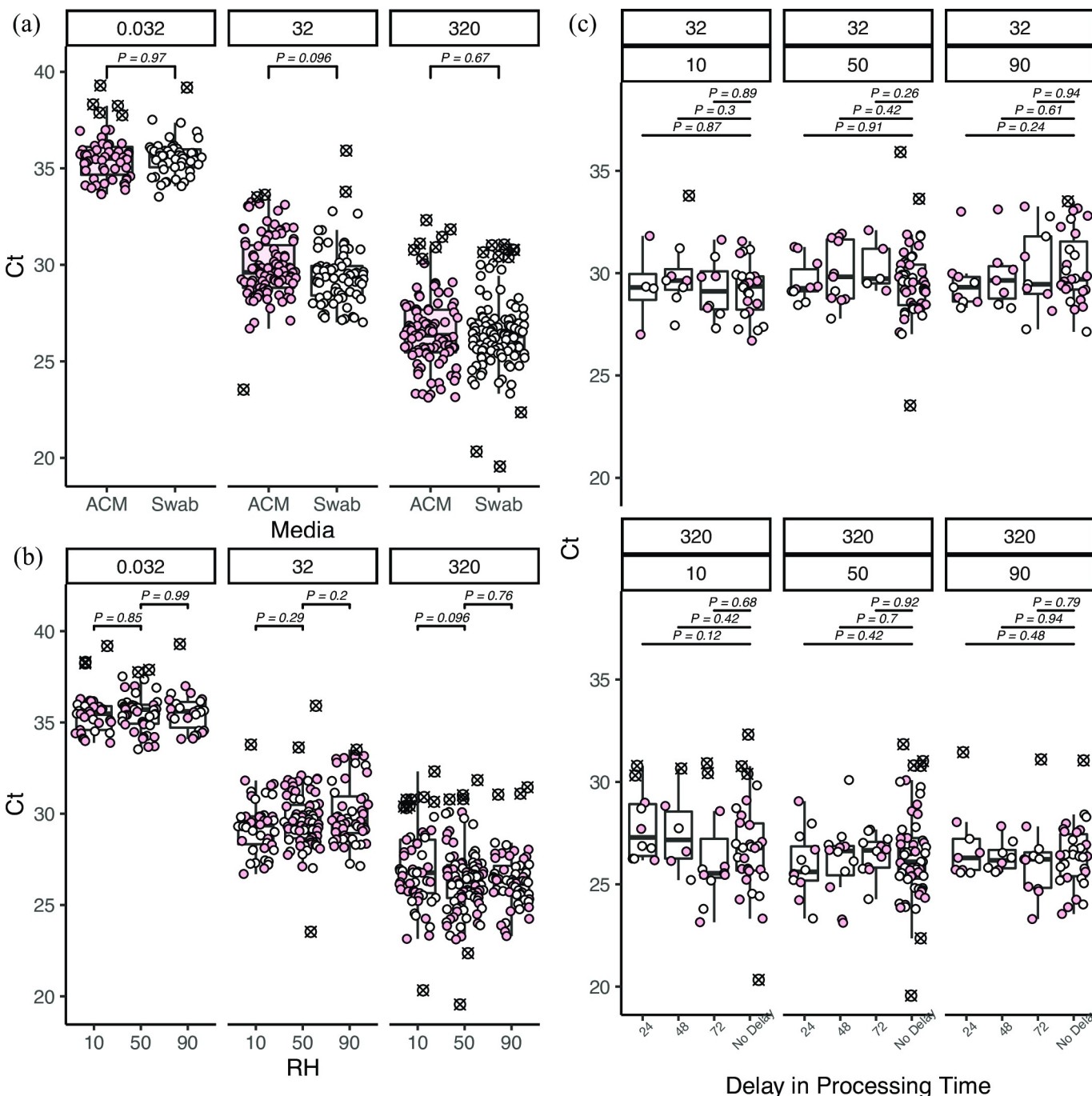

**Fig 5. _Bench series 3_ results.** Pink points represent samples collected from ACM and white points are samples collected from flocked swabs. Outliers are shown as crossed out circles (a) Measured Ct values at three main concentrations (0.032, 32, and 320 gc/L respectively) tested recovered from each capture media. (b) Measured Ct values at three measured aerosolized virus concentrations and RH levels. Outliers are shown as crossed out circles (c) Measured Ct values from samples processed immediately, with a 24-hour delay, with a 48-hour delay, and with a 72-hour delay. The top panel box is the aerosol concentration tested (32 gc/L or 320 gc/L) and the second panel box is the humidity at which the particles were aerosolized and at which the samples were maintained until being processed.

In the bench-scale aerosol experiments, the goal was to assess the ability of the AerosolSense sampler and the selected capture media to detect aerosolized heat-inactivated SARS-CoV-2 in a controlled environment. The published limit of detection of the TaqPath assay is ten genome

**Table 6. Percent of media tested positive for the presence of SARS-CoV-2 RNA expressed as genome copied dosed per L of room air irrespective of time.**

| 75 Minute Trials—1 ACH | | | |
|---|---|---|---|
| Low Dust | | High Dust | |
| Estimated Genome Copies / L | Percent Positive | Estimated Genome Copies / L | Percent Positive |
| 0.318 | 25% (1/4) | 0.318 | 0% (0/4) |
| 3.18 | 0% (0/4) | 3.18 | 50% (2/4) |
| 31.8 | 100% (4/4) | 31.8 | 75% (3/4) |
| 318 | 100% (4/4) | 318 | 100% (4/4) |
| 3,180 | ❁ | 3,180 | 100% (4/4) |
| **75 Minute Trials—6 ACH** | | | |
| Low Dust | | High Dust | |
| Estimated Genome Copies / L | Percent Positive | Estimated Genome Copies / L | Percent Positive |
| 0.197 | 25% (1/4) | 0.197 | 0% (0/4) |
| 1.97 | 25% (1/4) | 1.97 | 50% (2/4) |
| 19.7 | 25% (1/4) | 19.7 | 75% (3/4) |
| 197 | 100% (4/4) | 197 | 100% (4/4) |
| 1,970 | 100% (4/4) | 1,970 | 100% (4/4) |
| **8+ Hour Trials—0.8 ACH** | | | |
| Estimated Genome Copies / L | | Estimated Percent Positive | |
| 0.089 | | 100% (8/8) | |
| 0.89 | | 100% (8/8) | |
| 8.9 | | 100% (8/8) | |

Samples were considered positive if two out of the three genomic targets (N, S, ORF1ab) returned a Ct value ≤35. ❁ denotes that the concentration was not tested in that condition.

copy equivalents [41]. Limited detection began at an aerosol concentration of 0.032 gc/L (estimated 16 genome copies) and consistent detection (≥50%) was observed at 32 gc/L (estimated 16,000 total genome copies). Consistent detection of the RNA was likely not observed until well above the published assay limit of detection due to deposition of the virus on the surfaces within the glovebox, inefficiencies in the retrieval of eluate from the capture media, and potential inefficiencies during the RNA extraction process [52]. However, significant detection was observed at 32 gc/L, a similar aerosol concentration as has been observed previously in SARS-CoV-2- healthcare environments [1, 5, 43].

In order to assess two potential scenarios of aerosol surveillance in the real-world, the sampling protocol was tested against varying levels of RH and delayed processing time to mimic potential shipping or an inadvertent delay in sample collection or processing. In bench series trials, we observed no significant difference in measured Ct value of samples collected at different RH levels at any of the three different aerosol concentrations tested (Fig 2B). While previous research has demonstrated that coronavirus infectivity and survival can be significantly impacted by differing humidity levels [53–56], our results were in the context of 1) already inactive RNA, and 2) a very short aerosolization and sampling duration and distance. Since the total sampling period was maintained for 5 minutes at a very short distance (~30 cm), it is possible that any changes in RH may have required greater travel distance longer and potentially more than 5 minutes for deposition-related effects to have been observed. Similarly, delayed sample processing up to 72 hours was not found to have a statistically significant impact on the ability to detect SARS-CoV-2 RNA (Fig 2C). While the CDC recommends clinical samples remain under refrigeration at 4°C if they will not be processed immediately [57],

SARS-CoV-2 RNA has been demonstrated to show limited degradation up to a week after collection at room temperature as long as the viral envelope remains intact [58] and has been detected weeks after deposition in some cases [59]. The bench series results demonstrated consistent detection at and above previously measured aerosol concentrations in healthcare environments. Furthermore, the results indicate robustness against degradation after collection on the capture media at a range of RH values and processing delays. Therefore, additional experimental trials were conducted at the scale of a full room.

Respiratory aerosols typically range in diameter from 0.1–100 μm [60] and are typically categorized into two distinct categories: (i) coarse respiratory particles (> 5μm) and (ii) fine respiratory particles (≤5μm) [60]. Although respiratory aerosols produced by individuals range in diameter from 0.1–100μm, laboratory simulation experiments [61, 62], SARS-CoV-2 RNA detection [1, 63–66], and infectious aerosol sample collection [67] have demonstrated that the majority of SARS-CoV-2 infectious and potentially infectious particles are found in aerosols with diameters ≤5μm (i.e. fine respiratory particles). The aerosols created by the nebulizers in the previously described experiments, while not perfectly mimicking the total range of human respiratory output, emitted the majority of viral particles in the highly relevant fine respiratory particle range (Table 3). The AerosolSense Air Sampler has been rated to collect aerosol particles with diameters between 0.1 and 15μm [42]. Even though this collection range also does not encompass the entirety of human respiratory output, it could potentially collect, at a minimum, at least 85% of the available aerosolized RNA (assuming perfect collection efficiency). An important factor in determining if the sampler can collect the aerosol is whether or not the particle of a given size may reach the sampler inlet. We would expect that the AerosolSense could also collect larger particles as long as the particles can make it to the sampler inlet. The times for three particle sizes, 15μm, 20μm, and 30μm, to settle 1 meter in still air are 145 seconds, 82 seconds, and 30 seconds, respectively [68, 69; Full equations, including a spreadsheet for calculations, can be found in the supplementary material]. This is the vertical velocity of the particles. In the time the particles drop 1 meter, they will have traveled 1.116 meters, 0.66 meter, and 0.24 meter horizontally for the 15μm, 20μm, and 30μm particles with a horizontal velocity of 0.008 m/sec (supplementary material). For the 20μm and 30μm particles, by the time it takes for the particles to travel 1 meter, they will have fallen nearly 2 meters or more, sufficient time to drop out of the air. This travel distance illustrates that unless the release is very close to the inlet of the sampler, it is highly unlikely these particles will travel a sufficient horizontal distance to be able to be collected by the sampler. This is similar to a human subject; unless the release of these large particles is very close to the breathing zone, they are not likely to be inhaled by the subject. Taken together, we conclude that the AerosolSense will reliably collect the majority of aerosolized SARS-CoV-2 particles that are present within an enclosed space.

No statistical difference was observed in SARS-CoV-2 detection when impacted on flocked swabs or ACM (*P = 0.661*). Although flocked swabs are currently the standard for the collection of clinical samples for most respiratory tract viruses [25–27, 29, 51, 57], media similar to ACM has previously demonstrated superiority over flocked swabs for the sampling and detection of RNA viruses [50, 51]. Both materials reliably detected SARS-CoV-2 in aerosols and were easily integrated into a typical molecular workflow. However, due to their popularity in clinical diagnostics, flocked swabs may prove more difficult to obtain in times of extreme demand, as has been documented throughout the progression of the COVID-19 pandemic [70–72].

Room-scale aerosol trials demonstrated that detection was ≥50% at an estimated aerosol concentration of 31.8 gc/L when sampling took place for 75 minutes. However, when sampling took place for longer durations, the detection rate of SARS-CoV-2, even at an estimated

aerosol concentration as low as 0.089 gc/L, increased to 100%. The amount of dust present in the room, ranging from ambient dust to high dust load did not have a significant impact on the ability of the sampler to detect SARS-CoV-2 in aerosols (Table 4; *P = 0.308*). Additionally, the distance of the sampler from the aerosolization source (4' or 14') and the dust source did not have a significant impact on the ability to detect aerosolized SARS-CoV-2 or the intensity of the detected SARS-CoV-2 (*P = 0.622*).

Future investigations with the AerosolSense sampler should focus on the utility of bioaerosol sampling in real-world scenarios, both with and without the known presence of positive COVID-19 individuals. When positive aerosol samples are identified, epidemiologic investigations may benefit from the genomes recovered and the potential linking with clinical diagnostic genomes [73, 74]. Additionally, the sampler and sampling methods outlined throughout this manuscript could provide utility beyond SARS-CoV-2 surveillance and could be beneficial for the surveillance of other aerosol respiratory pathogens such as Influenza virus, Respiratory Syncytial virus (RSV), other coronaviruses, adenoviruses, and Rubeola virus as well as other pathogens of interest in healthcare and research settings such as antibiotic resistant organisms, *Mycobacterium tuberculosis*, and *Clostridioides difficile*.

## Conclusion

The results presented above demonstrate the utility of the AerosolSense sampler to detect SARS-CoV-2 RNA when paired with either the AerosolSense Capture Media or flocked swabs. Viral detection in aerosols was found to be consistent and reproducible when tested in a laboratory bench-scale setting and in a full-scale built environment. Aerosol SARS-CoV-2 detection was found to be robust against high levels of household dust, even at low estimated viral concentrations. Consistent detection of SARS-CoV-2 was achieved at estimated aerosol concentrations consistent with currently published aerosol concentrations found in healthcare settings [1, 5, 43] and detection was not significantly impacted at higher levels of air changes per hour or across different humidity levels and processing durations. All together, these results provide strong evidence for the utility of the AerosolSense sampler as an environmental surveillance tool for airborne pathogens like SARS-CoV-2 in a wide-range of indoor public spaces.

## Supporting information

**S1 Data.**
(XLS)

**S2 Data.**
(XLSX)

**S1 File.**
(PDF)

**S2 File.**
(PDF)

## Acknowledgments

The authors would like to thank the tireless team at BioBE for all of their help, including Georgia MacCrone, Liliana Barnatan, Vincent Moore, and Surbhi Nahata. The authors would like to thank Dr. Matthew Taylor at Montana State University for his guidance and expertise and the JRL-BSL3 for the contribution of heat-inactivated SARS-CoV-2. The authors would like to

thank Chuck Williams and Tony Schaffer, University of Oregon, for their aid in forming a relationship with Thermo Fisher Scientific. The authors would like to thank Siqi Tan, Arunava Dutta, and Geoffrey Gonzalez for their review of this manuscript and Jeff Ambs for his guidance concerning aerosol particle dynamics, settling time, and travel distance.

## Author Contributions

**Conceptualization:** Patrick Finn Horve, Leslie Dietz, Dale Northcutt, Jason Stenson, Kevin Van Den Wymelenberg.

**Data curation:** Patrick Finn Horve, Jason Stenson.

**Formal analysis:** Patrick Finn Horve.

**Funding acquisition:** Kevin Van Den Wymelenberg.

**Investigation:** Patrick Finn Horve, Dale Northcutt, Jason Stenson, Kevin Van Den Wymelenberg.

**Methodology:** Patrick Finn Horve, Jason Stenson, Kevin Van Den Wymelenberg.

**Project administration:** Patrick Finn Horve, Leslie Dietz, Kevin Van Den Wymelenberg.

**Supervision:** Leslie Dietz, Kevin Van Den Wymelenberg.

**Validation:** Patrick Finn Horve.

**Visualization:** Patrick Finn Horve.

**Writing – original draft:** Patrick Finn Horve, Kevin Van Den Wymelenberg.

**Writing – review & editing:** Patrick Finn Horve, Leslie Dietz, Dale Northcutt, Jason Stenson, Kevin Van Den Wymelenberg.

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
