## [Decision Letter · Decision Letter 0]

30 Jul 2021

PONE-D-21-14944

Evaluation of a Bioaerosol Sampler for Indoor Environmental Surveillance of Severe Acute Respiratory Syndrome Coronavirus 2

PLOS ONE

Dear Dr. Van den Wymelenberg

Thank you for submitting your manuscript to PLOS ONE. After careful consideration, we feel that it has merit but does not fully meet PLOS ONE’s publication criteria as it currently stands. Therefore, we invite you to submit a revised version of the manuscript that addresses the points raised during the review process.

In full agreement with the reviewer there are several important issues;

Firstly ; authors should provide a detailed description of the aerosols generated by the system and according to particle size monitoring explain why such aerosols resemble that generated by a human upon breathing.

Secondly. An accurate description of the particles present in the aerosols experimentally generated; how close are the particle ranges in the in vitro generated aerosols and in that generated by a human.

Thirdly. Once infected by Cov2 people might be hospitalized and kept in rooms that benefit from constant air filtration. Under these conditions are aerosols exhaled by a human detected and for how long?

We look forward to receiving your revised manuscript.

Kind regards,

Jean-Luc EPH Darlix, MG, Ph.D.

Academic Editor

PLOS ONE

2. Thank you for stating the following in the Competing Interests/Financial Disclosure* (delete as necessary) section:

“I have read the journal's policy and the authors of this manuscript have the following competing interests: Van Den Wymelenberg has a company called Duktile through which he provides healthy building consulting, including consulting related to viral pathogens, and he serves as a scientific advisor to EnviralTech, a company that conducts viral environmental surveillance, including in senior care facilities.”   

We note that one or more of the authors are employed by a commercial company: name of commercial company.

Additional Editor Comments (if provided):

Reviewers' comments:

Reviewer's Responses to Questions

**Comments to the Author**

1. Is the manuscript technically sound, and do the data support the conclusions?

Reviewer #1: Partly

2. Has the statistical analysis been performed appropriately and rigorously? 

Reviewer #1: Yes

3. Have the authors made all data underlying the findings in their manuscript fully available?

Reviewer #1: Yes

4. Is the manuscript presented in an intelligible fashion and written in standard English?

Reviewer #1: Yes

5. Review Comments to the Author

Reviewer #1: The authors describe the testing of a commercially available impaction-based bioaerosol sampling system to detect SARS-CoV-2. They first evaluated specific collection media and their use in typical lab procedures required to extract virus. They identified two media that withstood processing and provided consistent measurements in terms of cycle threshold (Ct). They then performed bench and room scale aerosol collection experiments using a Thermo Scientific AerosolSense Sampler. Investigators cultured and heat deactivated SARS-CoV-2 and nebulized it at concentrations from 0.0032 – 320 genome copies/L air in the glove box. In a room scale experiment, they aerosolized solutions to generate aerosols with 3.2-32000 genome copies/L air. The also investigated the effects of humidity, delayed sample processing, and high dust loads on collection efficiency.

My first issue with the manuscript is that there is very little detail presented on the aerosol generation systems used and no justification that these provide an accurate depiction of the aerosols generated by a human. There’s some amount of literature on the characteristics of exhaled aerosols and as the authors describe in the introduction. Their sizes range from submicron through 100’s of microns. My concern is that the authors may have used something akin to a medical nebulizer which produces aerosol only within a narrow range (apx. 1-10 microns). Information on the aerosol sizes generated for both the room and glove box experiments needs to be added to the methods and compared to the aerosol sizes generated by exhaled aerosols from people. Associated limitations for the study should be added to the discussion.

My second issue is that the manuscript doesn’t report the aerosol size range detectable by the sampler. From the product manual this range seems to be 0.1-15 microns in diameter. (https://assets.fishersci.com/TFS-Assets/CAD/Flyers/aerosolsense-faq.pdf ). The problem there is that’s only a small fraction of the aerosol sizes that might be generated by a person and that this device may provide a false sense of security because it may miss aerosols generated in larger aerosol size ranges which could contain substantially more virus that aerosols of the order of 1-10 microns. These aerosols would have shorter settling times, but still could circulate within a room. Information on the detectable aerosol size ranges for the collector should be reported in the methods. This should be compared to the aerosol sizes generated by exhaled aerosols from people. Associated limitations for the study should be added to the discussion.

Minor:

Methods: “SARS-CoV-2 deposited by CDC…” Is deposited the correct word?

Methods: Define Ct at first use

Methods: Describe the aerosol sizes generated by the bench and room nebulizer systems.

Results: In the studies media authors should define which end of their Likert scale is best and which is worst. This information should also be included in the caption of Table 3

6. PLOS authors have the option to publish the peer review history of their article (what does this mean?). If published, this will include your full peer review and any attached files.

Reviewer #1: **Yes: **Tim Corcoran

---

## [Author Response · Author response to Decision Letter 0]

25 Aug 2021

Editor Comments

Firstly ; authors should provide a detailed description of the aerosols generated by the system and according to particle size monitoring explain why such aerosols resemble that generated by a human upon breathing.

We agree that this is important information needed in order to properly contextualize the results that we have reported. We have added nebulization characteristics reported by Harvard Apparatus for the jet nebulizer, and we carried out some additional aerosolization experiments in order to characterize the particle distribution for the BLAM nebulizers. These characterizations through our own experiments and manufacturer information are available in the methods section (Table 2) as well as some interpretation in the discussion section. We have added an explanation about why these nebulized particle sizes are relevant to human respiratory particles to the discussion section.

Secondly. An accurate description of the particles present in the aerosols experimentally generated; how close are the particle ranges in the in vitro generated aerosols and in that generated by a human.

We have significantly expanded our discussion section concerning the particle sizes that were created by the utilized nebulizers, the detection range of the AerosolSense sampler and the particle settling dynamics of the particles of interest. 

Thirdly. Once infected by Cov2 people might be hospitalized and kept in rooms that benefit from constant air filtration. Under these conditions are aerosols exhaled by a human detected and for how long?

We thank the reviewer for their suggestion on a potential scenario where the sampling device could be deployed. We agree that the active air sampler has significant potential for the use in healthcare environments, however, we believe a discussion on this scenario is out of scope for this paper. We have a current preprint around this exact topic (https://www.medrxiv.org/content/10.1101/2021.03.26.21254416v1.full-text).

As you will find in the above preprint, there are many factors that would impact the ability of the AerosolSense sampler to detect SARS-CoV-2 RNA in a healthcare environment. Due to the multitude of factors that could potentially impact the presence of RNA within a healthcare environment (e.g. the patient viral dynamics, presence/absence of symptoms, breathing/intubation status, hospital ventilation strategies, hospital HVAC configuration, hospital cleaning protocols, etc…), we feel that it would be inappropriate to attempt to declare when detection of SARS-CoV-2 RNA would cease in such a generalized case. The main intent of our submitted manuscript was to characterize the AerosolSense sampler in a controlled environment. We believe that expanding beyond this more than we already have (final paragraph of the discussion) expands beyond the scope of this particular manuscript. We agree that the AerosolSense has potential for use in healthcare environments but we must respectfully decline to further address the detection of exhaled SARS-CoV-2 particles in healthcare environments as suggested. 

Reviewer Comments: 

Reviewer #1: The authors describe the testing of a commercially available impaction-based bioaerosol sampling system to detect SARS-CoV-2. They first evaluated specific collection media and their use in typical lab procedures required to extract virus. They identified two media that withstood processing and provided consistent measurements in terms of cycle threshold (Ct). They then performed bench and room scale aerosol collection experiments using a Thermo Scientific AerosolSense Sampler. Investigators cultured and heat deactivated SARS-CoV-2 and nebulized it at concentrations from 0.0032 – 320 genome copies/L air in the glove box. In a room scale experiment, they aerosolized solutions to generate aerosols with 3.2-32000 genome copies/L air. The also investigated the effects of humidity, delayed sample processing, and high dust loads on collection efficiency.

We would like to thank the reviewer for their very thoughtful and thorough review of our manuscript. Briefly, we have performed some additional trials in order to characterize the particle output of the nebulizers utilized in the experiments (summarized in table 2). We have also provided additional context and references as to the typical aerosol generation from COVID positive individuals. 

My first issue with the manuscript is that there is very little detail presented on the aerosol generation systems used and no justification that these provide an accurate depiction of the aerosols generated by a human. There’s some amount of literature on the characteristics of exhaled aerosols and as the authors describe in the introduction. Their sizes range from submicron through 100’s of microns. My concern is that the authors may have used something akin to a medical nebulizer which produces aerosol only within a narrow range (apx. 1-10 microns). Information on the aerosol sizes generated for both the room and glove box experiments needs to be added to the methods and compared to the aerosol sizes generated by exhaled aerosols from people. Associated limitations for the study should be added to the discussion.

We agree that this is important information needed in order to properly contextualize the results that we have reported. We have carried out some additional aerosolization experiments in order to characterize the particle distribution from the nebulizers that were utilized in the study. These characterizations through our own experiments and manufacturer information are available in the methods section (Table 2) as well as some interpretation in the discussion section. 

My second issue is that the manuscript doesn’t report the aerosol size range detectable by the sampler. From the product manual this range seems to be 0.1-15 microns in diameter. (https://assets.fishersci.com/TFS-Assets/CAD/Flyers/aerosolsense-faq.pdf ). The problem there is that’s only a small fraction of the aerosol sizes that might be generated by a person and that this device may provide a false sense of security because it may miss aerosols generated in larger aerosol size ranges which could contain substantially more virus that aerosols of the order of 1-10 microns. These aerosols would have shorter settling times, but still could circulate within a room. Information on the detectable aerosol size ranges for the collector should be reported in the methods. This should be compared to the aerosol sizes generated by exhaled aerosols from people. Associated limitations for the study should be added to the discussion.

We thank the reviewer for this suggestion on how to further discuss our results and, more importantly, the context of our results. We have significantly expanded our discussion section concerning the particle sizes that were created by the utilized nebulizers, the detection range of the AerosolSense sampler and the particle settling dynamics of the particles of interest. 

Minor:

Methods: “SARS-CoV-2 deposited by CDC…” Is deposited the correct word?

This is the suggested way to reference this material on the BEI resources website (https://www.beiresources.org/Catalog/animalviruses/NR-52281.aspx) where the original cultures were obtained. The full citation section of the resources reads: 

“Acknowledgment for publications should read “The following reagent was deposited by the Centers for Disease Control and Prevention and obtained through BEI Resources, NIAID, NIH: SARS-Related Coronavirus 2, Isolate USA-WA1/2020, NR-52281.””

Methods: Define Ct at first use

This has been corrected. 

Methods: Describe the aerosol sizes generated by the bench and room nebulizer systems.

This has been addressed in the methods and discussion sections. See previous comments. 

Results: In the studies media authors should define which end of their Likert scale is best and which is worst. This information should also be included in the caption of Table 3

We thank the author for this suggestion. This has been added.

---

## [Editor Report · Decision Letter 1]

8 Sep 2021

Evaluation of a Bioaerosol Sampler for Indoor Environmental Surveillance of Severe Acute Respiratory Syndrome Coronavirus 2

PONE-D-21-14944R1

Dear Dr. Van Den Wymelenberg

We’re pleased to inform you that your manuscript has been judged scientifically suitable for publication and will be formally accepted for publication once it meets all outstanding technical requirements.

Kind regards,

Jean-Luc EPH Darlix, MG, Ph.D.

Academic Editor

PLOS ONE
---

## [Editor Report · Acceptance letter]

5 Nov 2021

PONE-D-21-14944R1 

Evaluation of a Bioaerosol Sampler for Indoor Environmental Surveillance of Severe Acute Respiratory Syndrome Coronavirus 2 

Dear Dr. Van Den Wymelenberg:

I'm pleased to inform you that your manuscript has been deemed suitable for publication in PLOS ONE. Congratulations! Your manuscript is now with our production department. 

Kind regards, 

on behalf of

Professor Jean-Luc EPH Darlix 

Academic Editor

PLOS ONE